# Airway Surface Liquid pH Regulation in Airway Epithelium Current Understandings and Gaps in Knowledge

**DOI:** 10.3390/ijms22073384

**Published:** 2021-03-25

**Authors:** Miroslaw Zajac, Elise Dreano, Aurelie Edwards, Gabrielle Planelles, Isabelle Sermet-Gaudelus

**Affiliations:** 1Department of Physics and Biophysics, Institute of Biology, Warsaw University of Life Sciences, 02-776 Warsaw, Poland; miroslaw_zajac@sggw.edu.pl; 2Institut Necker Enfants Malades, INSERM U1151, 75015 Paris, France; elise.dreano@inserm.fr; 3Centre de Recherche des Cordeliers, Sorbonne Université, INSERM, Université de Paris, 75006 Paris, France; gabrielle.planelles@inserm.fr; 4Department of Biomedical Engineering, Boston University, Boston, MA 02215, USA; aued@bu.edu; 5Laboratoire de Physiologie rénale et Tubulopathies, CNRS ERL 8228, 75006 Paris, France; 6Centre de Référence Maladies Rares, Mucoviscidose et Maladies de CFTR, Hôpital Necker Enfants Malades, 75015 Paris, France; 7Clinical Trial Network, European Cystic Fibrosis Society, BT2 Belfast, Ireland; 8European Respiratory Network Lung, 75006 Paris, France

**Keywords:** pH, CFTR, SLC26A4, ATP12A, lung

## Abstract

Knowledge on the mechanisms of acid and base secretion in airways has progressed recently. The aim of this review is to summarize the known mechanisms of airway surface liquid (ASL) pH regulation and their implication in lung diseases. Normal ASL is slightly acidic relative to the interstitium, and defects in ASL pH regulation are associated with various respiratory diseases, such as cystic fibrosis. Basolateral bicarbonate (HCO_3_^−^) entry occurs via the electrogenic, coupled transport of sodium (Na^+^) and HCO_3_^−^, and, together with carbonic anhydrase enzymatic activity, provides HCO_3_^−^ for apical secretion. The latter mainly involves CFTR, the apical chloride/bicarbonate exchanger pendrin and paracellular transport. Proton (H^+^) secretion into ASL is crucial to maintain its relative acidity compared to the blood. This is enabled by H^+^ apical secretion, mainly involving H^+^/K^+^ ATPase and vacuolar H^+^-ATPase that carry H^+^ against the electrochemical potential gradient. Paracellular HCO_3_^−^ transport, the direction of which depends on the ASL pH value, acts as an ASL protective buffering mechanism. How the transepithelial transport of H^+^ and HCO_3_^−^ is coordinated to tightly regulate ASL pH remains poorly understood, and should be the focus of new studies.

The airway epithelium is central to the defenses of the lung. It constitutes an interface between the internal milieu and the external environment, acting as a barrier against particles deposited in the larger airways, and inactivating infectious microorganisms in the lower airways [1]. The luminal side of airway epithelia is lined by a thin layer of fluid called airway surface liquid (ASL) [2,3]. ASL arises mainly from submucosal gland secretions and transepithelial hydro-osmotic movements. Its composition is finely tuned, more specifically its pH. Increasing data have provided evidence that normal ASL is slightly acidic relative to the interstitium, and that defects in its pH regulation are associated with various respiratory diseases [4]. Research on the mechanisms of acid and base secretion in airways has progressed significantly recently. The aim of this review was to summarize the known mechanisms of ASL pH regulation in airway physiology and their implication in lung diseases, as well as to highlight gaps in knowledge.

## 1. Methods

A literature review was performed according to PRISMA guidelines [5]. An exhaustive systematic review was performed based on the following keywords: airway surface liquid pH, bicarbonate, CFTR, SLC26A4, SLC26A9, shunt pathway, SLC4A4, ATP12A, carbonic anhydrase, SLC9A1, and V-ATPase. Experimental models and methods of pH assessment were specifically considered. The flow diagram of the studies we selected is shown in Figure 1.

## 2. Surface Liquid of the Airway Bronchial Epithelium, an Important Player in Airway Physiology

The airway bronchial epithelium is composed of a variety of cell types, described in Figure 2 [1,6].

In the large airways, the ciliated, goblet, and basal cells are predominant, whereas in the small airways, the secretory cells are found more frequently. Ciliated cells account for at least 50% of all epithelial cells within human airways, and are terminally differentiated. They express up to 300 cilia per cell, whose coordinated beating enables mucous clearance out of the airways. Goblet cells contain acidic-mucin granules. They can self-renew, as well as transdifferentiate into ciliated cells. Secretory (clara/club) cells secrete surfactants and specific antiproteases. Basal cells display stem cell-like properties and give rise to secretory and ciliated cells in response to epithelial injury. Ionocytes are a rare cell type, recently identified in the adult murine and human trachea, and in human proximal bronchi; they express cystic fibrosis transmembrane regulator (CFTR) protein at high levels [6,7]. Pulmonary neuroendocrine cells (PNEC) are ubiquitous in human adult airway epithelium, and located between epithelial cells adjacent to the basement membrane. Submucosal glands build invaginations throughout the cartilaginous airways in the trachea and large human airways. They are only present in the uppermost part of the mouse trachea. They are composed of mucous cells, serous cells, and myoepithelial cells. Thus, luminal mucus originates from both mucous and goblet cells. The two main mucins in human airways are MUC5AC, mainly produced in the goblet cells of the surface epithelium, and MUC5B, mainly produced in the mucous cells of the submucosal glands. MUC5AC is thought to be secreted as an acute response to environmental insults, while MUC5B is involved in the response to chronic infection and inflammation.

Regulation of ASL composition and mucociliary clearance is critical for normal airway function. The ASL is a ~10 µm bilayer made up of a periciliary liquid (PCL) and a mucus layer (MCL). The PCL is a watery layer which bathes the cilia, and is in direct contact with the epithelial cells. The MCL is a gel-like layer sitting over the tips of the cilia. The PCL allows the cilia to beat and strike at the underside of the MCL. Whereas the PCL is composed of 96% water, 1% salts, 1% lipids, 1% proteins, and 1% mucus [2,8], the MCL layer is a heterogeneous mixture of polypeptides and cellular debris tethered together on the PCL surface by MUC5AC and MUC5B complexes [9,10,11]. It is thought that the mucous layer acts as a fluid reservoir to maintain ASL hydration [12].

The ASL pH and bicarbonate (HCO_3_^−^) concentration are critical for several key functions of the airway epithelium.

At the cellular level, pH is involved in the regulation of ion transporters, and therefore in trans- and para-cellular salt and water movement and ASL homoeostasis. Indeed, the equilibrium between different redundant pathways for net chloride (Cl^−^) secretion and net sodium (Na^+^) absorption ensures the extremely precise regulation of ASL volume [13]. Chloride secretion mainly involves the cystic fibrosis transmembrane regulator protein (CFTR), both directly and indirectly, because CFTR also promotes the activity of other Cl^−^ transporters, including Cl^−^/HCO_3_^−^ exchangers and ANO1 (a Ca^2+^ activated Cl^−^ channel) [14]. Importantly, the luminal concentration of HCO_3_^−^ directly modulates the level of CFTR expression by stimulating soluble adenylate cyclase (sAC) and cAMP cellular production, leading to nuclear accumulation of the CREB transcription factor [15]. Similarly, ASL pH affects the membrane expression of ENaC, the main pathway for Na^+^ reabsorption, by regulating the activity of short palate lung and nasal epithelial clone 1 (SPLUNC1) antimicrobial peptide. This protein binds to αβγ-ENaC and causes the internalization of αγ-ENaC, thus preventing activation of the channel by serine proteases [16,17,18]. Importantly, this peptide fails to function at pH values below 7.0 due to pH-sensitive salt bridges, resulting in abnormal Na^+^ and water absorption [16,17].

At the epithelial level, pH and HCO_3_^−^ are crucial regulators of innate airway defense [19,20]. This is due, at least partially, to a reduction in the killing properties of antimicrobial peptides, such as LL-37, because their cationic electric charge is modified at pH values below 7 [21]. The physiological relevance of this observation has been demonstrated in vivo by showing that lowering the pH of lung porcine epithelium ASL from 7.58 to 6.35 decreased its bacterial killing capacity [22]. This combines with the intrinsic effects of HCO_3_^−^ on bacterial growth, airway colonization, and biofilm formation inhibition [23,24,25,26,27,28].

Finally, at the organ level, ASL pH modulates mucociliary and cough clearance, a process that enables continuous drainage of the airways, and thus proper air conduction. Mucociliary clearance is a complex phenomenon involving different factors: the mucus which traps exogenous particles and pathogens, the cilia whose beating moves up secretions, and the periciliary fluid which protects the epithelium against dehydration and bathes the cilia [29]. Mucus rheology and the viscoelastic properties of ASL are finely regulated by HCO_3_^−^. Indeed, mucins are condensed by calcium in secretory granules due to their acidic groups [30]. When they are secreted, they undergo expansion by as much as 1000-fold [30,31], triggered by the chelation of calcium by HCO_3_^−^ [32,33,34]. Additionally, pH values below 7.0 decrease ciliary beating in vitro (50% at pH = 5.5 and almost 100% at pH = 3.5) [35]. This is due to the fact that sAC activity is decreased at low intracellular HCO_3_^−^ concentrations (≤10 mM), thus decreasing local cAMP concentration necessary for cilia beating [36,37]. As a whole, defective ciliary beating and increased mucus viscosity combine to decrease mucociliary clearance (MCC) at acidic pH.

## 3. ASL pH in Physiology and Disease

### 3.1. Methods of Measurements

There are two classes of methods for measuring pH values.

Optical methods use the pH-dependent color changes of organic dye molecules (indicators) that are weak bases or acids. For the purpose of ASL pH measurements, the dextran-coupled (cell impermeant) pH fluorescent dyes BCECF, Fluorescein, HCC, or HPTS are employed, by adding them to the ASL directly [16,22,38,39,40,41].

The potentiometric methods measure the electrical voltage between a reference electrode and a pH electrode. Ionophore-filled home-made microelectrodes or commercial miniaturized pH electrodes have been used for determination of ASL pH in samples from human bronchial epithelial (HBE) cells [16,42,43]. However, these measurements in small samples are subjected to multiple confounding factors, including the immersion depth of the measuring electrode, placement of the reference electrode, disturbance of the epithelial layer, change in hygrometry, carbon dioxide concentration, and temperature in the vicinity of the measurement. Rapid developments in the field of sensors have enabled pH measurements in vivo, using mobidium pH probes [44,45,46], monocrystalline antimony catheters, or in-gold combined pH-glass electrodes [47]. More recently, Schultz et al. used a pH-sensitive luminescent dye-based fiber-optic probe. The dye was embedded in a hydrogel matrix, which limits interactions between the dye and ASL proteins that could potentially alter the measured pH values, and provides a high signal-to-noise ratio allowing for accurate measurements [48].

### 3.2. ASL pH Values in Physiological Conditions

The studies listed in Table 1, Table 2, Table 3 and Table 4 have shown that ASL pH in normal airways ranges in vivo between 5.6 and 6.7 in the nasal mucosa, and is around 7.0 in bronchia. Indeed, studies by McShane et al. showed that ASL pH was more alkaline in lower airways (7.1 ± 0.1) than in upper airways (6.6 ± 0.1) [47]. Similar ranges of values were observed in murine and porcine animal models. Such a wide range of ASL pH values may be explained by technical limitations. For example, obtaining stable recordings in vivo is challenging because of CO_2_ variations during breathing, which can skew pH measurements [48,49,50]. Moreover, applying an electrode to the epithelium immediately changes the transport equilibrium and therefore alters ASL pH.

To avoid problems with in vivo measurements, ASL pH may be measured in vitro. Most studies focused on HCO_3_^−^ transport have been performed in monolayers of Calu3, a cell line derived from a bronchial adenocarcinoma. Even though Calu3 cells are representative of submucosal cells, they are often considered to be a convenient model for epithelial airway cells, because they display strong cAMP-stimulated HCO_3_^−^ transport and they express CFTR. They are clearly different from other ciliated epithelial cell lines or native tissue cells (as listed in Table 2), since they display characteristics of both serous and mucus cells [78]. Fewer studies have been performed in ciliated epithelial cell lines or primary airway 2D cultures using fluorescent indicators or ion selective electrodes under controlled conditions (temperature, pCO_2_, and humidity) [58,68]. Interestingly, these studies show an ASL pH close to 7.4 in human bronchial primary cells. Whether these in vitro observations can translate to in vivo physiology is still unknown [22,50,58]. In vitro studies in primary bronchial cells also have a number of limitations, including the fact that the relative proportion of different cells may differ according to the samples, and transporter expression levels according to the culture medium, as recently pointed out by Saint Cricq et al. [79].

### 3.3. ASL pH Values in Disease

In 1930, Hilding et al. reported “acidic” nasal secretions (i.e., below the blood pH of 7.4) in inflammatory acute lung diseases, and “alkaline” pH in the common cold [80]. It was thus hypothesized that airway pH may be an indicator of airway disease, either as a cause or a consequence. Since then, various studies have shown changes in ASL pH in specific diseases. Inflammation and infection seem to raise pH above the values observed in healthy airways, including pneumonia (7.2–7.4) [81], rhinitis (7.2–8.3) [77], and chronic bronchitis (7.6–7.8) [76], with the exception of one study on active bacterial infection (5.6–6.2) [49].

Many studies have suggested decreased pH in the airways of asthmatic [82,83,84] and Chronic Obstructive Broncho Pulmonary Disease (COPD) patients [85,86]. This is mainly based on the observed reductions in the pH value of exhaled breath “condensate” compared to healthy controls. This may reflect more intense airways inflammation, but obviously cannot be extrapolated to ASL pH values [84,86].

The main relevant studies are summarized in Table 1, Table 2, Table 3 and Table 4. In vitro studies have shown that ASL pH is decreased in the context of *CFTR* mutations, as compared to epithelia carrying WT CFTR [8,22,42,43,50,51,52,53,54,55,56,57,58,59,60,61,62,63,64,65,66,67,68,69,70]. The majority of studies focused on the most frequent CFTR mutation, p. Phe508delCFTR (F508del). In 2011, Cho et al. observed that the rate of cAMP-dependent base secretion in CF nasal tissues was significantly lower than that in WT cells (11.8 ± 2.4 nmol min^−1^ cm^−2^, versus 57.2 ± 9.2 nmol min^−1^ cm^−2^) [87]. Interestingly, Coakley et al. showed in F508del cell lines exposed to a luminal acid challenge that the lack of HCO_3_^−^ secretion resulted in a defective re-alkalinization of ASL [42].

Few groups have focused on submucosal glands. Jayaraman et al. found that submucosal gland secretions in CF patients exhibited a pH of 6.97 ± 0.06 but high viscosity [51]. In contrast, Song et al. reported a pH that was 0.6 pH units higher in healthy subjects than in CF patients, corresponding to a three-fold increase in HCO_3_^−^ concentration [70]. This observation suggests that submucosal glands might contribute to ASL pH homeostasis.

In vivo data are discordant. Lower pH values were observed within the nasal epithelium of CF patients in comparison to healthy controls [44,47], while other studies did not report differences [48]. To date, the debate on ASL pH values in the context of CFTR mutations has not been resolved. If proved true, this abnormally low pH, or at least the defective buffering of acidic load, could explain a number of physiopathological aspects in lung disease, such as increased mucus thickness, defective innate defense, bacterial colonization, and the inflammation observed in CF epithelium [88,89].

The ASL pH value is the result of transepithelial acid and base transport and buffering power. The studies describing the main transporters involved in these processes are reported below.

## 4. Bicarbonate Transport in Airway Cells

Several important findings on the mechanisms underpinning HCO_3_^−^ transmembrane transport were obtained by Devor et al. in 1999 [90]. Measurements of short-circuit current and labeled fluxes in Calu3 cell lines provided convincing functional evidence that basolateral HCO_3_^−^ entry involves the electrogenic, coupled transport of Na^+^ and HCO_3_^−^, and that apical HCO_3_^−^ secretion involves CFTR and other transporters.

### 4.1. Apical HCO_3_^−^ Transport

#### 4.1.1. CFTR

CFTR is a channel that belongs to the ATP binding cassette (ABC) superfamily. The channel is gated by ATP binding/hydrolysis at its two transmembrane domains and by the phosphorylation of its large intracellular regulatory (R) domain. R phosphorylation is mostly achieved by the cAMP/PKA pathway (and less efficiently by the PKC phosphorylation pathway) [91]. Patch-clamp experiments in transfected cells have shown that CFTR exhibits a high anionic selectivity (P_Na_/P_Cl_ ~0.03) and a small linear conductance (~10 pS) [92]. Various anions may flow across the activated channel, down their favorable transmembrane electrochemical potential gradients [93]. Chloride is the favorite substrate among halide ions, as both its transmembrane electrochemical gradient and its permeability are the most favorable (the halide permeability sequence is Br^−^ > Cl^−^ > I^−^ > F^−^). Many other anions can be transported by CFTR, among them glutathione and HCO_3_^−^ (P_Cl_/P_HCO3_ ~ 4) [94,95]. This supports the role of CFTR, not only in Cl^−^ transport, but also in pH regulation.

CFTR is expressed at the apical membrane of airway ciliated epithelial cells, in submucosal glands, and in the recently described ionocytes [7,96,97]. Even though ionocyte-type cells are rare (1–2% of epithelial cells), single-cell RNA sequencing and Ussing chamber experiments in human bronchial epithelial cells (HBECs) demonstrated that they express 50% of *CFTR* transcripts, causing 60% of CFTR-mediated current, whereas the common ciliated epithelial cells, thought to be the major location of CFTR, express only 1.5% of *CFTR* transcripts and mediate 4% of the mean channel current [97]. However, the precise role of ionocytes in Cl^−^ secretion (and a fortiori in HCO_3_^−^ secretion) in the native epithelium remains unclear. Secretory cells might also play a predominant role in the expression and function of CFTR in the lungs, as recently reported [98].

At least three points increase the complexity of understanding Cl^−^ vs. HCO_3_^−^ transport by CFTR. First, CFTR is not only an anionic channel but also a regulator of other transport systems, including HCO_3_^−^ transporters [99,100,101]. Physical and/or functional interactions between CFTR and several transporters from the SLC26 family (A3, A6, A8, A9…) have been well documented [102,103,104]. It has been proposed that CFTR itself can act as a Cl^−^/HCO_3_^−^ exchanger [105], but this hypothesis is debated [106]. Second, the respective selectivity of the CFTR channel for Cl^−^ and HCO_3_^−^ appears to be dynamically regulated. The transport of Cl^−^ increases at intracellular pH (pH_i_) values of 6.6 and 6.3, whereas alkaline pH_i_ inhibits Cl^−^ flow [107]. Moreover, the amount of CFTR is rate-limiting for HCO_3_^−^ transport in contrast to Cl^−^ secretion, which nearly reaches the WT level in CFTR+/F508del pig epithelia [28]. In addition, a low intracellular Cl^−^ concentration may switch CFTR from a predominant Cl^−^ channel to a predominant HCO_3_^−^ channel, by activating intracellular Cl^−^-sensitive kinases (with-no-lysine kinase WNK1, oxidative stress-responsive kinase 1, OSR1, and sterile 20/SPS1-related proline/alanine-rich kinase, SPAK) [108,109]. Although this has been shown in pancreatic cells, this may be a mechanism that is also in place in the lung, as WNK1 and OSR1 are largely present in the lung [110]. Interestingly some *CFTR* mutations located in transmembrane domains selectively decrease the transport of HCO_3_^−^ by hindering the physical association between CFTR and WNK1 [109,111,112]. Third, HCO_3_^−^ transport is often assessed indirectly, i.e., by measuring pH changes in experimental protocols. It is in this context very difficult to differentiate between the effects of CFTR on pH values, and those of other transporters, which renders the interpretation of results difficult [113].

All these observations indicate the importance of CFTR-mediated HCO_3_^−^ transport in the pulmonary epithelium. This is reinforced by the recent observation that correctors of defective CFTR increase the permeability of the mutated channel to HCO_3_^−^ more than to halide ions [114].

#### 4.1.2. ANO1

Studies in HEK293 cells have shown that the Cl^−^ channel ANO1 becomes highly permeable to HCO_3_^−^ at high intracellular [Ca^2+^] [100]. Interestingly, in epithelial ciliated cells, a cross-activation between CFTR and ANO1 (TMEM16A) has been demonstrated, involving compartmentalized Ca^2+^ and cAMP crosstalk in specific plasma membrane domains containing GPCRs, CFTR, and TMEM16A [115].

#### 4.1.3. Apical Cl^−^/HCO_3_^−^ Exchangers

SLC26A4 and A9 belong to the solute carriers SLC26 transporter family, which is composed of a N-terminal transmembrane domain (TMD) connected to a C-terminal sulfate transporter anti-sigma factor antagonist (STAS) domain [116]. Multiple studies have shown physical, biochemical, and functional interactions between SLC26 anion exchangers and CFTR, leading to reciprocal activation [101,102,117]. This has especially been described for SLC26A3 and A6 [102,118], and more recently for SLC26A9 [103]. Interestingly, mutations in the STAS domain abolished this functional activation [102,103,119], suggesting a physical interaction between the STAS domain of the SLC26 transporters and the R domain of CFTR [102,117].

#### 4.1.4. SLC26A4 (Pendrin/PDS)

Pendrin is expressed preferentially in goblet cells, and at low levels in ciliated epithelial cells, but not in ionocytes [120]. Its expression at the apical membrane is increased by the pro-inflammatory cytokines IL-4, IL-13, and IL-17A [121,122]. The role of SLC26A4 in HCO_3_^−^ transport is still a matter of debate.

Garnett et al. showed that in Calu3 cells, increases in pendrin activity by cAMP/PKA can induce ASL alkalinization up to pH 7.9 (corresponding to a final concentration of 75 mM HCO_3_^−^) [101]. Pendrin remained active after the addition of GlyH-101 (a CFTR inhibitor) and basolateral DIDS (a Cl^−^ channel blocker), which suggests a mechanism independent of CFTR-mediated HCO_3_^−^ secretion. This might not be, however, physiological, as lower pH values are more frequently reported (Table 2).

Indeed, Shan et al. [62] reported lower HCO_3_^−^ concentrations in forskolin-stimulated fluid secretions (up to pH 7.55, corresponding to a final concentration of 31 mM), and a reduced HCO_3_^−^ flux after inhibition of CFTR, similarly to cells that do not express CFTR.

These observations are consistent with electroneutral Cl^−^/HCO_3_^−^ exchange by pendrin working in parallel with electrogenic Cl^−^ secretion mediated by CFTR [101,123]. In this model, the coupled Cl^−^/HCO_3_^−^ transport is mediated by the cAMP signaling pathway via protein phosphatase 1 (PP1), which inhibits HCO_3_^−^/Cl^−^ exchange by basolateral AE2 and activates CFTR, thereby enhancing the electrogenic efflux of Cl^−^, which is further recycled by SLC26A4, leading ultimately to HCO_3_^−^ secretion [101]. According to this model, CFTR acts mainly as a Cl^−^ transporter that fuels SLC26A4 to transport HCO_3_^−^ into the ASL. A similar mechanism has been described in pancreatic ductular cells between SLC26A6 and CFTR [101,124].

Several studies have supported the hypothesis of CFTR-pendrin coupling. In primary cultures of differentiated human airway epithelia, and in secretory cells where both proteins were co-expressed, Rehman et al. observed that the combination of TNFα+IL-17 increased CFTR and pendrin expression. This was associated with elevated ASL pH stemming from increased HCO_3_^−^ secretion [120]. Simonin et al. [58] found that inhibition of pendrin by a specific inhibitor decreased ASL pH. However, this is in contrast to Haggie et al. [67], who observed no effect on ASL pH. A possible explanation for these contradictory results is that pendrin expression may vary according to inflammation and culture conditions.

Indeed upregulation of pendrin transcription has been shown in chronic rhinosinusitis and in rodent models of inflammatory lung diseases [125,126,127]. More recently, Bajko et al. [128] showed that stimulation with interleukins IL-4, IL-13, or IL-17a increased pendrin levels in human bronchial epithelial cells from CF patients (CF HBECs) and non-CF donors (HBECs) [121,129]. Indeed, modulation of pendrin expression/activity by inflammation may represent a protective mechanism for re-alkalinization of ASL and epithelium defense during disease.

#### 4.1.5. SLC26A9

SLC26A9 is widely expressed in the luminal membranes of HBE cells, where it contributes to constitutive Cl^−^ secretion [130,131]. Patch clamp studies, on both cell lines and transfected HEK cells, showed that SLC26A9 is a highly selective Cl^−^ channel with linear current-voltage characteristics and minimal HCO_3_^−^ permeability [99,132]. Additionally, electrophysiological studies on different models (HEK, *Xenopuslaevis* oocytes, and animal models) showed that SLC26A9 may act as a Cl^−^/HCO_3_^−^ exchanger working in tandem with an apical Cl^−^ transporter, such as CFTR [133,134]. More speculatively, the channel might also switch its permeability from Cl^−^ to HCO_3_^−^, similarly to CFTR [108,135,136]. To date, the contribution of SLC26A9 to HCO_3_^−^ transport in airways remains unknown, and may be cell/tissue dependent.

### 4.2. Basolateral HCO_3_^−^ Transport

#### 4.2.1. Cl^−^/HCO_3_^−^ Exchanger (AE2)

The anion exchanger type 2 (AE2, or SLC4A2) is expressed at the basolateral membrane of airway epithelium and thought to participate in fluid secretion [137]. AE2 performs the reversible electroneutral exchange of Cl^−^ for HCO_3_^−^, as directed by the electrochemical potential gradients, and is in part responsible for the regulation of cytosolic pH and cell volume [138]. Its activity is downregulated by cAMP agonists, via a PKA-independent mechanism involving Ca^2+^, calmodulin, and the protein kinase CK2. This mechanism was confirmed in primary human nasal epithelia, where CK2 inhibition abolished the activity of AE2. In Calu3 cells, the involvement of AE2 in anion secretion [139,140] entails the coupled inhibition of AE2 and the activation of CFTR and pendrin by cAMP agonists, as described above [101].

#### 4.2.2. Electrogenic Na-Coupled Bicarbonate Co-Transport (NBCe, SLC4A4)

Na^+^-coupled bicarbonate symports belong to the large SLC4 family (see for review [141]). Among them, NBCe are electrogenic transporters, mediating the co-transport of 1 Na^+^ with 2 or 3 HCO_3_^−^, consequently transporting a net negative electrical charge. NBCe1-B and NBCe2 (aka pNBC1 and NBC4) were identified at the basolateral cell membrane of Calu3 cells [142], confirming an earlier conclusion that an electrogenic, inwardly directed coupled transport of Na^+^ and HCO_3_^−^ was the basolateral step of transcellular base secretion, intimately related to Cl^−^ and HCO_3_^−^ apical secretion [90]. It is noteworthy that the regulation of NBCe1-B is finely coupled to that of CFTR [143,144]. First, NBCe1-B has a consensus phosphorylation site for PKA located in the N-terminus and is stimulated by cAMP. This is in line with the forskolin-induced stimulation of net transepithelial HCO_3_^−^ transport in Calu3 cells [90]. Second, phosphorylated IRBIT (Inositol-1,4,5-trisphosphate receptor-binding protein released with IP_3_), a known activator of CFTR, also stimulates the activity of NBCe1-B [145], and antagonizes the inhibition of both NBCe1-B and CFTR by the WNK-SPAK pathway [108,146]. Finally, in transfected HeLa cells, NBCe1-B is sensitive to the intracellular Cl^−^ concentration [147]. Further studies are needed to confirm the above regulations in other cell models.

#### 4.2.3. Carbonic Anhydrases (CA)

Carbonic anhydrases (cytosolic, mitochondrial, or membrane-bound CAs) catalyze the hydration of CO_2_ and H_2_O into carbonic acid (H_2_CO_3_), which immediately equilibrates with HCO_3_^−^ and H^+^, thereby leading to a quasi-immediate equilibrium between CO_2_ and HCO_3_^−^. In the lungs, CA activity was demonstrated thanks to the effect of acetazolamide on the conversion of H^14^CO_3_^−^ to ^14^CO2 [148]. The two fastest CAs, i.e., the cytosolic CAII and the membrane-bound CAIV (which may also contribute to cytosolic CA activity) [149], are expressed in the lung. It has been proposed that the association of CAII and CA IV to various acid–base transporters (AE1, NHE1, NBCe1) forms a metabolon that greatly accelerates the transcellular transport of acid equivalents [150,151,152,153]. Such a functional complex in Calu3 cells is supported by the acetazolamide-induced decrease in HCO_3_^−^ secretion [90,154], which suggests that some fraction of the transported HCO_3_^−^ results from metabolic cell production. At the ASL interface, if CA catalyzes the immediate equilibrium of CO_2_ with HCO_3_^−^ and H^+^, large and rapid pH variations should be observed at each inspiration/expiration [155]. However, only slow and acetazolamide-insensitive pH variations were measured, indicating minimal CA activity in ASL, thus preventing large and rapid changes of ASL pH upon CO_2_ variations during inspiration/expiration [156]. This means that a disequilibrium pH (i.e., a discrepancy between the measured pH, and the pH value if CO_2_-HCO_3_^−^-H^+^ reactions were at equilibrium) may prevail in the ASL. This implies that one can infer HCO_3_^−^ concentrations from measured pH values only in steady-state conditions, an important matter of concern during laboratory experiments, and most importantly, in vivo.

Several arguments suggest that CA expression/function plays a role in CF. First, CA IV targeting may be altered in CF cells [157]. Second, the genetic loss of CA XII function is associated with CF-like features [158]. Third, in co-cultured Calu3 cells/WT murine lymphocytes, CAII and IV are up-regulated, and HCO_3_^−^ secretion is enhanced, whereas this defense mechanism against infection is impaired in Calu3/CF lymphocytes co-culture [159].

### 4.3. Paracellular HCO_3_^−^ Transport

The paracellular pathway has mostly been explored to assess the transport of medicinal drugs and various molecules (such as cytokines or allergens) in the pulmonary epithelium [160,161]. Few studies have focused on the ionic permeation properties of the shunt barrier, which depend largely on the expressed claudins [162]. In human airway primary cultures, a preferential cationic selectivity was evidenced by measurements of dilution potentials under open circuit conditions [163]. Recent studies confirmed this selectivity, and showed that the permeabilities of the shunt pathway to Cl^−^ and HCO_3_^−^ are equal, but that their conductances differ due to their different luminal concentrations [164]. According to the authors’ calculations, the paracellular HCO_3_^−^ flux is secretory under basal conditions (ASL pH < basolateral pH), but it reverses to absorption when ASL pH is >7.0, a condition reached when transcellular HCO_3_^−^ secretion is enhanced by the presence of pro-inflammatory cytokines [120,164].

## 5. Acid Transport in Airway Cells

Different observations point to H^+^ secretion into ASL. Even though both transcellular and paracellular routes across the airway epithelium are permeable to HCO_3_^−^, normal steady-state ASL pH has been found to be below blood pH. Moreover, ASL pH re-acidifies at an initial rate of ~0.2 pH/hour after mild alkalinization, which is equivalent to a net acid secretion of 3.5 nmol h^−1^ cm^−2^ (as per the calculation of Fisher et al. in a ASL of volume 2.5 µL cm^−2^ and buffer capacity of 7 mM/UpH) [165]. We describe below the main transporters involved in acid transport.

### 5.1. Basolateral Secretion: Na^+^/H^+^ Exchangers

Na^+^/H^+^ exchangers form a family of transporters that regulate pH homeostasis, with nine known members: NHE1–9 (SLC9A1 to A9). NHEs mediate the entry into the cell of extracellular Na^+^ in exchange for cytosolic H^+^, with a 1:1 stoichiometry, as dictated by the chemical gradients of Na^+^ and H^+^ [166]. Al-Bazzaz et al. showed by transcript expression that NHE1 is expressed along the entire human respiratory tract. In particular, the relative abundance of NHE1 mRNA was found to be higher in the trachea and distal airways in humans [167].

At the functional level, Paradiso et al. observed a Na^+^-dependent re-alkalization upon cytosolic acidification (NH_4_Cl prepulse method); the pH_i_ recovery was blocked by serosal but not mucosal amiloride in human nasal epithelial cell monolayers [168]. This suggests a basolateral localization of NHE1.

Other studies, however, have suggested the potential localization of a Na^+^/H^+^ exchanger isoform at the apical membrane, based on decreased acid secretion after treatment with mucosal amiloride in primary human airway epithelia, intact distal airways from pigs, and tracheal tubes from sheep [56,169,170,171]. This potential NHE-mediated H^+^ secretion into the ASL might explain the association between *SLC9A3* variants and the susceptibility to airway infections in CF patients, but this hypothesis needs experimental support [172,173,174,175]. As a whole, the localization of NHEs, their role in regulating ASL pH, and their clinical relevance need to be further investigated.

### 5.2. Apical Secretion

#### 5.2.1. H^+^/K^+^ ATPase

HKA2 (ATP12A) belongs to the P2-type ATPase family and shares sequence homologies with both the gastric H,K-ATPase (ATP4A) and the Na,K-ATPase (ATP1A). HKA2 mediates the electroneutral exchange of H^+^ for potassium (K^+^) but may also function in a Na^+^/K^+^ exchange mode [176,177,178]. HKA2 is inhibited by ouabain (previously thought to be specific to the Na,K-ATPase) but its sensitivity to SCH28080 (a potent inhibitor of the gastric H,K-ATPase) is debated. HKA2 is located on the apical side of respiratory epithelia, particularly in the goblet cells [68,179]. Its trafficking to the apical cell membrane is triggered by its association with the β subunit of Na,K-ATPase, ATP1B1, which is increased in inflammatory conditions, whereas the Na,K-ATPase (composed of a αcatalytic subunit associated with ATP1B1) remains strictly located to the basolateral cell membrane [68]. Of note, HKA2 expression is low in murine airways, and this may partly explain the very mild pulmonary CF phenotype in this species [40]. In human primary cell cultures and fresh tissue, the contribution of ATP12A to ASL pH was thoroughly investigated by Coakley et al. [42]. They measured a K^+^-dependent, ouabain-sensitive, SCH28080-insensitive ASL acidification, a result that supports an acidifying process mediated by HKA2. However, even though ASL pH was lower in CF cultures than in WT cultures, the acidification rate was not different. The authors concluded that HCO_3_^−^ secretion mediated by CFTR buffers HKA2-mediated H^+^ secretion, but that the activity of this pump is the same in CF and WT cells. This thus would explain the “hyper acidity” of the CF ASL [42]. Interestingly, similar to CFTR, HKA2 is upregulated by an increase in intracellular cAMP [180]. Supporting the role of HKA2 in the lower ASL pH observed in CF airway epithelia, Simonin et al. showed that using ouabain to inhibit HKA2 restored the pH value of CF ASL to that of WT [58]. The deleterious involvement of HKA2-induced acidification of ASL during infectious episodes was demonstrated by Shah et al. in pigs and humans. Shah et al. concluded that in non-CF epithelia, HCO_3_^−^ secretion by CFTR balanced H^+^ secretion by HKA2, but that in CF epithelia, unbalanced proton secretion occurs, and impairs the host defenses against bacteria [28]. This observation is reinforced by the upregulation of HKA2 by several interleukins in the context of inflammation [68]. Inhibition of HKA2 in lungs is therefore expected to be beneficial during CF, as it might increase ASL pH. Up to now, only ouabain has been proven to inhibit HKA2, but its toxicity precludes its use for therapy.

#### 5.2.2. Hydrogen Voltage-Gated Channel 1 (HVCN1)

The fundamental function of voltage-gated H^+^ channels is acid extrusion from the cells. Their open probability depends only on pH or membrane depolarization [181]. H^+^ channels were found in airway epithelia, both in primary and immortalized cell lines, and their activity was localized to the apical membrane [171,182,183]. Ivovannisci et al. demonstrated that the HVCN1 channel contributes to ASL pH by opening its transmembrane pore when extracellular pH exceeds 7.0 [184]. Since the membrane potential of airway epithelia is stable, H^+^ channel gating must be primarily regulated by the pH gradient across the apical membrane (increased extracellular pH rather than decreased intracellular pH). This gradient may be enhanced by DUOX oxidase (nicotinamide adenine dinucleotide phosphate-oxidase) [182] and/or mitochondria tightly packed at the apical membrane of airways [185], which can contribute to intracellular H^+^ production. Fischer proposed a model with a passive feedback system controlling ASL pH, which combines CFTR alkalizing the ASL by secreting HCO_3_^−^ and voltage-gated H^+^ channels secreting H^+^ [186]. It might be disrupted in CF, as suggested by a recent study showing that HVCN1 protein levels in lysates of nasal cells were significantly lower in CF patients than in healthy subjects [187].

#### 5.2.3. Vacuolar H^+^-ATPase

Vacuolar (V-type) H^+^-ATPases play an important role in regulating pH, by pumping H^+^ across membranes against the pH gradient, using the energy from ATP hydrolysis [188]. The contribution of V-ATPase to ASL pH remains unclear. Whereas some groups observed that bafilomycin, a specific V-ATPase blocker, raises ASL pH in Calu3 cells and in distal pig bronchi [56,62,189], others found that bafilomycin has minimal or no effects in human, pig, and cow airway epithelia [70,171,182,190]. Of note, a study on primary nasal epithelial cells and cultured lung showed that *Pseudomonas aeruginosa* inactivates V-ATPase, thereby reducing the expression and trafficking of CFTR [191].

## 6. Proposed Model

Altogether, the studies described above indicate that the airways constitutively secrete both acid and base at substantial rates, which concur to tightly regulate ASL pH. The model shown in Figure 3 recapitulates our current understanding of acid–base transport across airway epithelia. Apical HCO_3_^−^ secretion is mediated by CFTR, pendrin, and possibly ANO1, and may be facilitated by the coupling between Cl^−^ secretion via CFTR (and possibly other Cl^−^ channels) and Cl^−^ recycling through Cl^−^/HCO_3_^−^ exchangers. Apical H^+^ secretion is mediated by ATP-driven H^+^ transporters, such as ATP12A and V-ATPase, that carry H^+^ against the electrochemical potential gradient. In contrast, HVCN1-mediated H^+^ secretion is passive, and can only occur when the electrochemical potential H^+^ gradient is reversed.

Basolateral NBC activity coupled with CA-mediated HCO_3_^−^ formation provide HCO_3_^−^ for apical secretion. The protons that are concomitantly formed are reabsorbed by basolateral NHE1, or secreted into the ASL. In the basal state, AE2, acting in concert with Na^+^-HCO_3_^−^ cotransporters, enables basolateral Cl^−^ loading and HCO_3_^−^ recycling, which reduces the electrochemical driving force for HCO_3_^−^ transport across the apical membrane. Under cAMP-stimulated conditions, AE2 is inhibited whereas NBCe1 is activated, both of which enhance the driving force for apical HCO_3_^−^ secretion. Paracellular HCO_3_^−^ transport, the direction of which depends on the ASL pH value, acts as an ASL protective buffering mechanism.

## 7. Current Gaps in Knowledge

Many aspects of the model proposed above remain to be elucidated, including the contribution of each epithelial cell type, the role of specific transporters, and regulatory pathways.

Airway secretion is a very complex biofluid, containing ASL, as well as mucins, cell micro-organisms, and their byproducts.

Mucins are acidic with a low isoelectric point, and should therefore contribute to low pH buffering, an additional protective factor in inflammatory disease, where their amount is greatly increased. Indeed, mucins have an excellent acid buffer capacity. This has been well documented in the intestine [192]. In the lung, Holma et al. studied induced sputum from smokers, and determined a pH of 6.8 with a buffering capacity by around 7 µM/pH unit of sputum; they also found that the patients whose sputum contained less mucin were more prone to pollution induced lung damage [193,194]. A more recent study on sub mucosal gland (SMG) secretions from the Verkman laboratory yielded a relatively acidic pH of 7.0–7.2 [65]. Finally, Song, et al. reported in human gland secretions induced by pilocarpine a buffer capacity of 12 mM/pH unit, decreasing to ~3.7 mM/pH unit, in HCO_3_^−^ free conditions, thereby confirming that HCO_3_^−^ is a substantial contributor to mucus pH [70].

Those observations are of the utmost importance. Indeed, in vitro measurements in excised porcine bronchi suggested that airway fluid originates primarily from the SMGs [195]. Moreover, in vitro optical studies from individual SMGs from pigs and ferrets showed that cAMP agonists induced a combination of Cl^−^ and HCO_3_^−^ secretion in SMG fluid/mucus secretion [196]. This process was reduced in CF ferrets and pigs [197]. This is consistent with the known high level of CFTR expression in SMG serous cells [198] and the fact that CFTR-Inh 172 inhibits pilocarpine- and forskolin-induced airway SMG secretion in WT, but not CF pigs and humans [57]. The relative contribution of SMGs, goblet cells, and epithelial ciliated cells to ASL pH buffering at basal state and during acid load is still unclear. This is, at least partly, because most of the mechanistic studies were done on specific cell lines and not on full tissue explant, which would allow better dissecting of the different pathways of H^+^ and HCO_3_^−^ secretion, and better understanding the physiopathology of diseases that alter ASL pH. The pH of ASL is also intimately related to its cellular and bacterial content. Pulmonary diseases can cause accumulation of inflammatory cells, which in turn release reactive oxygen species and other byproducts of aerobic metabolism that cause damage to lipids, proteins, and DNA, and modify the buffering capacity of airway secretions. Pulmonary diseases also induce microbiota dysbiosis, as well as alterations of the structure of the airways and the formation of niches, that favor the growth and increase of common commensal bacteria. The relationship between microbiota and pH is complex. Indeed, Ratske et al. [199] showed that, bacteria can change the pH of the microenvironment by producing metabolites, namely indoles and organic acids. Organic acids lower lung mucus pH, which in turn favors anaerobic fermentation and the production of propionic acid. In the lung, this could contribute to a further decrease in the buffering capacity of airway secretions and low sputum pH [200].

Our understanding of the function of each cell type remains very limited. Which cells contribute significantly to HCO_3_^−^ and/or H^+^ secretion? Do ciliated cells play a role in the process of pH homeostasis? What is the role of secretory cells? Does this change along the airways? Is substantial acid and base secretion made possible by large H^+^ and HCO_3_^−^ fluxes across a small number of cells, or by small fluxes across a large number of cells?

The respective roles of the different transporters, and how their action is coordinated are still unclear. The following issues can be highlighted.

Regarding HCO_3_^−^ transport, the relative contribution of CFTR and pendrin to apical HCO_3_^−^ secretion remains unclear. Some investigators have reported that CFTR is the main, if not the only, apical HCO_3_^−^ transporter in Calu3 and native airway epithelial cells [62,201]. However, several recent studies have suggested that pendrin-mediated HCO_3_^−^ secretion plays a more important role than CFTR-mediated HCO_3_^−^ secretion in regulating ASL pH, as observed in porcine airway epithelial cells [50], human nasal, and bronchial epithelial cells pre-treated with IL-4 [129], and cAMP-activated Calu3 cells [101]. Recently, SNPs in the SLC26A9 gene were found to be associated with CF-related disease onset, suggesting that SLC26A9 acts as a CFTR regulator [202], and may be involved in the response to CFTR-directed therapeutics [172,203]. Whether this might involve modification of pH ASL is unknown.

The mechanisms underlying basolateral HCO_3_^−^ transport have not been fully deciphered. NBCe1-B and NBCe2 have been identified on the basolateral side of the pulmonary epithelium. Both are electrogenic, but whether they operate as a 1Na^+^:2HCO_3_^−^, 1Na^+^:3 HCO_3_^−^, or 1Na^+^:1HCO_3_^−^:1CO_3_^2−^ cotransporter is not yet established [141,204]. Aside from a heuristic viewpoint, the stoichiometry of these symports has physiological relevance. Depending on the value of the basolateral membrane potential and their respective reversal potential, the direction of the NBC-mediated transport may reverse, switching from basolateral HCO_3_^−^ influx into the cell, to efflux out of the cell.

Regarding H^+^ transport, the physiological role of HKA2 and its therapeutic target potential remain to be clarified. Whether apical K^+^ recycling via K^+^ channels acts to enhance H^+^ secretion by HKA2 is unknown. Indeed, upregulation of the K^+^ channel KCNMB4a by IL-4 has been shown by microarray analysis in bronchial epithelial cells (BE37), but this was not characterized at the protein level [179]. Interestingly, Coakley et al. showed that the K^+^ channel blocker barium did not alter ASL acidification rates, arguing against a significant K^+^ conductance on the apical cell membrane [42]. Moreover, several studies indicate that HKA2 might switch from a H^+^/K^+^-ATPase to a Na^+^/K^+^-ATPase [176,178]. In the native renal collecting duct, this mode leads to luminal Na^+^ secretion [205]. Interestingly, a recent study showed that the Na^+^/K^+^ exchange mode of HKA2 is induced by increases in intracellular cGMP [206], and thus indirectly related to NO, whose level is decreased in CF epithelial cells. Importantly, there have been conflicting findings regarding the sensitivity of HKA2 to the inhibitors of the gastric H/K-ATPase (see for review [207]). These discrepancies are possibly explained by a tissue- (or species-) dependence of HKA2 sensitivity. The search for non-toxic HKA2 inhibitors is under active investigation.

Finally, as described above, the binding of CAII and CAIV to acid–base transporters may form a “metabolon” [150,151,153]. As reviewed by Pushkin and Kurtz, this functional complex would greatly enhance transcellular HCO_3_^−^ fluxes across transporters from the SLC4 family [152], but this remains a matter of debate [208]. Further studies on the physical/functional link between CA and acid–base transporters in the different cell types in normal and in CF conditions are necessary, and may identify novel therapeutic targets.

## 8. Conclusions

There is increasing evidence that ASL pH is linked to airway function, and that the tight regulation of ASL pH is crucial to airway homeostasis. Recent progress has led to a better understanding of acid and base transporters in airway epithelium. However, knowledge on how the transepithelial transport of H^+^ and HCO_3_^−^ is coordinated to tightly regulate ASL pH is clearly limited. This should be the focus of new studies to improve our understanding of physiology and disease, and identify therapeutic targets.

## Figures and Tables

**Figure 1 ijms-22-03384-f001:**
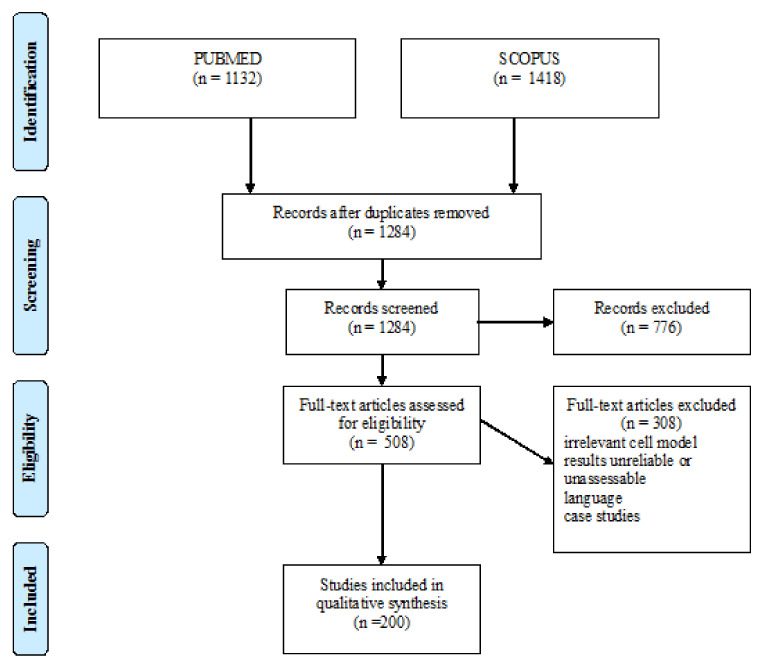
PRISMA flow diagram.

**Figure 2 ijms-22-03384-f002:**
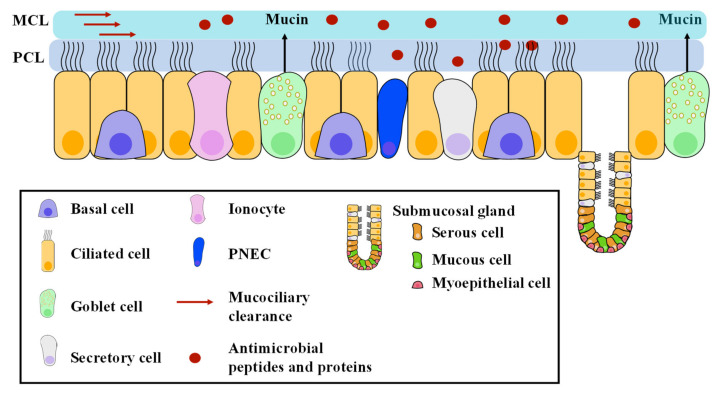
Pseudo-stratified airway epithelium and physiological role of main cells.

**Figure 3 ijms-22-03384-f003:**
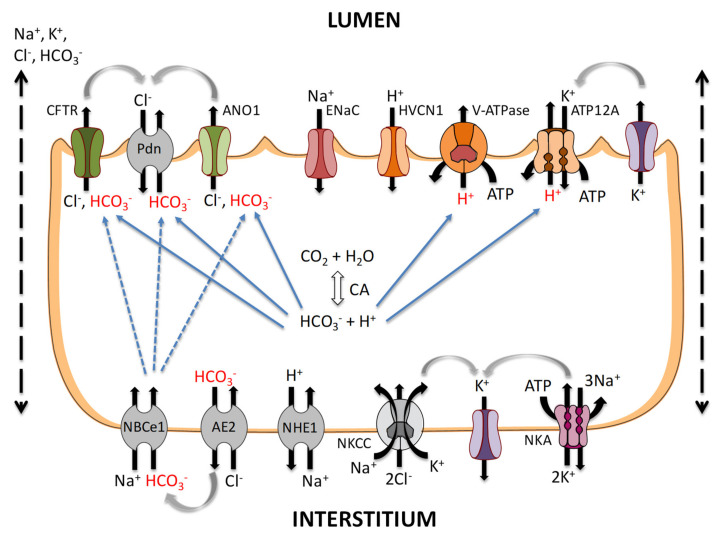
Distribution of the currently known acid and base transporters of airway epithelial cells.

**Table 1 ijms-22-03384-t001:** Airway surface liquid (ASL) pH in animal models.

Sample/Model	pH Value		Method Reference	
	WT	CF		
Mouse	6.95 ± 0.03	6.84 ± 0.07	BCECF-dextran	[51]
	7.14 ± 0.01 (in vivo)		BCECF-dextran	[52]
	6.98 ± 0.16		pH electrode	[50]
	7.28		BCECF-dextran	[8]
Rat	7.25 ± 0.05	6.42 ± 0.12	Not specified	[53]
Ferret	6.84 ± 0.03		pH electrode	[54]
Rabbit *	6.92 ± 0.01		pH electrode	[55]
Cow	6.81 ± 0.04 (25 mM bicarbonate)		BCECF-dextran	[52]
	6.98 ± 0.05 (no bicarbonate)		BCECF-dextran	[52]
Pig	7.14 ± 0.04	6.94 ± 0.05	Optode (in vivo)	[22]
Primary Bronchial epithelia	7.37 ± 0.05	7.05 ± 0.03	SNARF pH indicator	[22]
	6.93 ± 0.04		pH electrode	[56]
	~7.1	7.2	pH electrode	[50]
Gland fluid	6.9 ± 0.06		BCECF-dextran	[57]

* alveolar subphase; WT: Wild Type; CF: Cystic Fibrosis.

**Table 2 ijms-22-03384-t002:** ASL pH in vitro measurements of human samples.

Sample/Model	pH Value		Method	Reference
	WT	CF		
Cell lines				
CFBE41σ	7.42 ± 0.02 *	7.15 ± 0.01 *	pH electrode	[58]
		7.24 **	X Ray microanalysis	[59]
16HBE14 σ	7.14 ± 0.02		pH electrode	[60]
	7.16 **		pH electrode	[59]
Calu3	~7.2		BCECF-dextran	[61]
	7.55 ± 0.04	7.28 ± 0.02	pH electrode	[62]
NuLi-1/CuFi-1	7.52 ± 0.07	6.88 ± 0.02	SNARF pH indicator	[63]
C38/IB3-1/	7.32 ± 0.08	7.02 ± 0.04	pH electrode	[64]
Primary cells (bronchi)				
	6.81 ± 0.20		BCECF-dextran	[65]
	6.6 ± 0.1		SNARF-1	[66]
	ΔpH = −0.096 ± 0.029 *	ΔpH = −0.146 ± 0.011 *	pH electrode	[42]
	~7.4	~7.1	BCECF-dextran	[67]
	7.77	7.31	pH electrode	[68]
	7.35 ± 0.09		pH electrode	[43]
	7.43 ± 0.06 *	7.26 ± 0.02 *	pH electrode	[58]
	7.35 ± 0.05	6.70 ± 0.03	pH electrode	[69]
Submucosal gland secretions				
	7.18 ± 0.06	6.57 ± 0.09	BCECF-dextran	[70]
	6.97 ± 0.06		BCECF-dextran	[65]

* after 6 h Ringer incubation; ** after 3 h Ringer incubation; WT: Wild Type; CF: Cystic Fibrosis.

**Table 3 ijms-22-03384-t003:** ASL pH in vivo measurements of human samples in healthy controls and patients with cystic fibrosis.

Sample/Model	pH Value		Method	Reference
	WT	CF		
Nose				
Edge of nostril/adults	5.5 ± 0.1	5.6 ± 0.1	Monocrystalline antimony catheter	[47]
4 cm from nares/adults	6.7 ± 0.13	6.2 ± 0.1	Monocrystalline antimony catheter	[47]
	6.6 ± 0.1	6.8 ± 0.10	Gold probe	[47]
		5 to 7.2 *	Mobidium pH probe	[44]
Neonates	6.4 ± 0.2	5.2 ± 0.3 (4.5–6.9)	Mobidium pH probe	[44,45]
Lower airway/children				
	7.1 ± 0.1	7.1 ± 0.2	Gold probe	[47]
	7.00 ± 0.12	6.98 ± 0.15	Fiberoptic probe	[48]

*: different genotypes; WT: Wild Type; CF: Cystic Fibrosis.

**Table 4 ijms-22-03384-t004:** ASL pH in vivo measurements of human samples in patients with diseases other than cystic fibrosis.

	pH Value	Method	Reference
Pneumonia	6.62 ± 0.07	pH electrode	[71]
	6.72	pH electrode	[72]
Chronic lung diseases	6.64 ± 0.08	pH electrode	[71]
COPD	6.21 ± 0.37	pH test strip	[73]
Acute exacerbation of COPD (AECOPD)	6.89 ± 0.53	pH test strip	[73]
Chronic rhinosinusitis	6.7 ± 0.6	pH electrode	[74]
Pulmonary tuberculosis (sputum)	7.00 (range 5.50–8.37)	pH electrode	[75]
Chronic bronchitis	7.59 (mucoid)	pH electrode	[76]
	7.83 (purulent)	pH electrode	[76]
Rhinitis	7.2–8.3	pH electrode	[77]

COPD: Chronic Obstructive Broncho Pulmonary Disease.

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
