# Peer review of "Airway Surface Liquid pH Regulation in Airway Epithelium Current Understandings and Gaps in Knowledge"

_ijms, 2021, doi:10.3390/ijms22073384_

Round 1

Reviewer 1 Report

A group of international reviewers have produced a comprehensive literature of mechanisms of airway surface liquid (ASL) pH regulation and their possible impacts on selective airway diseases. Focus is appropriately directly at airway cellular transport, processes responsible for acid and base secretion with mentions of select regulation processes. The paper is well presented, references for the most part are current and updated and figures and tables are appropriate. The review is useful and timely. I have only limited comments to make for the authors to consider in minor revision.

Comments

  1. Page 6, first full paragraph, more mucus (increased mucus thickness) should result in better buffering of an acid load as mucus glycoproteins represent an excellent acid buffer (think of the stomach) as well as an excellent antioxidant and antacid functions (these are related) could be better developed as this is a very important constituent of airway surface fluid. This reader disagrees with the comment on page 14, second full paragraph, “the contribution of mucus to the buffer capacity of the ASL has never been investigated”. Mucus in both the g-I tract and respiratory tract is a very strong acid buffer well documented in the literature.
  2. It is worth mentioning the the potential confounding effects of ASL cellular constituents to affect ASL pH.  Both disease inflammatory cells and the respiratory tract microbiome potentially represent significant contributors.

Reviewer 2 Report

The review is very well written and covers many things. 

I think in the present the review is accepted.

No comments.

Author Response

thank you for your positive comments